# PEAβ Triggers Cognitive Decline and Amyloid Burden in a Novel Mouse Model of Alzheimer’s Disease

**DOI:** 10.3390/ijms22137062

**Published:** 2021-06-30

**Authors:** Luana Cristina Camargo, Michael Schöneck, Nivethini Sangarapillai, Dominik Honold, N. Jon Shah, Karl-Josef Langen, Dieter Willbold, Janine Kutzsche, Sarah Schemmert, Antje Willuweit

**Affiliations:** 1Institute of Biological Information Processing, Structural Biochemistry (IBI-7), Forschungszentrum Jülich, 52425 Jülich, Germany; l.camargo@fz-juelich.de (L.C.C.); d.honold@fz-juelich.de (D.H.); d.willbold@fz-juelich.de (D.W.); j.kutzsche@fz-juelich.de (J.K.); s.schemmert@fz-juelich.de (S.S.); 2Institut für Physikalische Biologie, Heinrich-Heine-Universität Düsseldorf, 40225 Düsseldorf, Germany; 3Institute of Neuroscience and Medicine, Medical Imaging Physics (INM-4), Forschungszentrum Jülich, 52425 Jülich, Germany; m.schoeneck@fz-juelich.de (M.S.); nivethini.sangarapillai@uni-marburg.de (N.S.); n.j.shah@fz-juelich.de (N.J.S.); k.j.langen@fz-juelich.de (K.-J.L.); 4JARA-Brain-Translational Medicine, JARA Institute Molecular Neuroscience and Neuroimaging, 52062 Aachen, Germany; 5Department of Neurology, RWTH Aachen University, 52062 Aachen, Germany; 6Department of Nuclear Medicine, RWTH Aachen University, 52062 Aachen, Germany

**Keywords:** transgenic mice, behavioral tests, amyloid plaques, Alzheimer´s disease, mouse model, amyloidosis, neurodegeneration, neuritic plaques, neuroinflammation, cognitive decline

## Abstract

Understanding the physiopathology of Alzheimer’s disease (AD) has improved substantially based on studies of mouse models mimicking at least one aspect of the disease. Many transgenic lines have been established, leading to amyloidosis but lacking neurodegeneration. The aim of the current study was to generate a novel mouse model that develops neuritic plaques containing the aggressive pyroglutamate modified amyloid-β (pEAβ) species in the brain. The TAPS line was developed by intercrossing of the pEAβ-producing TBA2.1 mice with the plaque-developing line APPswe/PS1ΔE9. The phenotype of the new mouse line was characterized using immunostaining, and different cognitive and general behavioral tests. In comparison to the parental lines, TAPS animals developed an earlier onset of pathology and increased plaque load, including striatal pEAβ-positive neuritic plaques, and enhanced neuroinflammation. In addition to abnormalities in general behavior, locomotion, and exploratory behavior, TAPS mice displayed cognitive deficits in a variety of tests that were most pronounced in the fear conditioning paradigm and in spatial learning in comparison to the parental lines. In conclusion, the combination of a pEAβ- and a plaque-developing mouse model led to an accelerated amyloid pathology and cognitive decline in TAPS mice, qualifying this line as a novel amyloidosis model for future studies.

## 1. Introduction

Dementia is the most common form of neurodegenerative diseases, leading to a decline in cognitive functions over time. In 2020, an estimated amount of around 27 million people in the world lived with dementia [1,2,3]. Among the neurodegenerative dementias, Alzheimer’s disease (AD) has the highest prevalence with approximately 60% of all dementia cases [4,5]. Besides the cognitive decline, two main pathological hallmarks occur in AD brains, most of which are known as neurofibrillary tangles and neuritic plaques. The latter ones are mainly composed of different aggregated species of the amyloid-β (Aβ) peptide. Aβ aggregation is initiated by the interaction of Aβ monomers, which assemble into toxic Aβ oligomers. Further aggregation leads to Aβ fibrils that are the main component of the neuritic plaques and can be found extracellularly in human AD brains. Following the amyloidogenic pathway, Aβ monomers are produced by the cleavage of the amyloid precursor protein (APP) [6] through cleavage by the β-secretase, instead of the α-secretase in the non-amyloidogenic pathway, and the consecutive cleavage by the γ-secretase [7]. A minor amount of AD cases are classified as familial AD (fAD), in which different gene mutations have been identified to be the primary cause of AD. Most of the fAD mutations are discovered in the *APP* gene, as well as in an active component of the γ-secretase complex, the presenilin-1 or -2 genes (*PSEN1*, *PSEN2*) [8]. Mutations in these genes lead to increased processing of APP in the amyloidogenic pathway and consecutively to the secretion of Aβ. Taking advantage of this, a number of fAD mutations have been used to generate transgenic animal models developing AD-like amyloidosis in the brain for studying AD pathophysiology and testing of new therapeutic options.

A widely used fAD mutation for transgenic mouse models is the so-called APPswedish mutation (APPswe), which occurs by the change of amino acid residues outside the Aβ domain, facilitating Aβ production [6,9,10]. The combination of an APPswe transgene with a second one, harboring a mutated presenilin transgene, has been shown to additionally increase formation of the Aβ1-42 species, which is highly prone to aggregation and, thus, leads to a progressive amyloidosis phenotype in the brain of double transgenic mice [11]. A double transgenic mouse model commonly used in AD research is the APPswe/PS1ΔE9 (APP/PS1) line, which carries both an APPswe and a mutated presenilin-1 (PS1) transgene. The mice start to develop amyloid plaques and neuroinflammation within six months, which progressively intensifies with age [8,12,13,14]. Despite the presence of abundant neuritic plaques later in life, the mice do not develop obvious neurodegeneration [14]. Nevertheless, cognitive deficits have been described for this line, although with variable onset [15,16]. One study described a deficit in spatial learning and memory in the so-called Morris water maze (MWM) paradigm with 15 months of age [17]. All in all, this line is a very common mouse model whose amyloidosis seems to be very similar to the human disease, in particular due to the presence of neuritic plaques in both cortex and hippocampus. However, the APP/PS1 line has been shown to be not particularly suited for amyloid imaging with radiotracers routinely used in the clinic, as studies by Stenzel et al. [18] and our research group suggested. A difference in plaque morphology or composition has been discussed as the cause for lower binding of amyloid radiotracers to plaques of this mouse model [19]. In addition, the presence of truncated Aβ species has been proposed to be important for amyloid radiotracer binding and AD-like plaque morphology [20].

A variety of different Aβ species can be detected in amyloid plaques of AD patients. While the main Aβ variants are Aβ1-40 and Aβ1-42, a significant proportion consists also of N-terminally truncated species [21]. Studies identified that N-terminally truncated and pyroglutamate-modified Aβ (pEAβ) represents up to 25% in neuritic plaques in human AD patients´ brains [22,23,24]. The pEAβ is formed in a post-translational modification of exposed glutamate by the glutaminyl cyclase in both glutamate position 3 (pEAβ3-42) and 11 (pEAβ11-42) of truncated Aβ [25]. These modifications change the overall charge of the Aβ peptide, which increases its hydrophobicity and, thus, facilitates Aβ aggregation and prevention of degradation by proteases [26]. Besides that, pEAβ is more neurotoxic than other Aβ species (for a review, see [27]).

The relevance of pEAβ for AD pathophysiology has also been demonstrated in transgenic mouse models. Firstly described by Alexandru et al. in 2011 [28], the TBA2.1 mouse model was generated to secrete pEAβ3-42 in the brain, which intracellularly forms small Aβ aggregates along with a strong neurodegeneration and severe motor deficits in homozygous mice. However, unlike most of the other amyloidosis mouse lines, the TBA2.1 mice do not develop neuritic plaques but only smaller Aβ particles deposited in the brain [29].

Development of an AD-like amyloidosis phenotype, including neuritic plaques and neuroinflammation, could be shown in a variety of transgenic mouse lines. Additionally, behavioral deficits, in particular cognitive impairment in learning and memory, were described to be present in most lines (for a review, see [30]). However, neurodegeneration, which is a typical feature of human AD, was only observed in a few transgenic amyloidosis models, including the TBA2.1 line [31]. In an attempt to generate an improved mouse model harboring a combination of AD-relevant hallmarks, i.e., the aggressive phenotype produced by pEAβ, an abundant formation of neuritic plaques and extensive cognitive decline, the novel TAPS mouse line was generated. This line was created by cross-breeding of heterozygous APP/PS1 and TBA2.1 mice and the phenotype of the resulting triple transgenic mice was followed over a period of 20 months in comparison to the parental lines. As a result, we demonstrate that by addition of pEAβ the amyloid pathology is further accelerated, with earlier onset and increased deposition of neuritic plaques in the brain. Furthermore, the TAPS mice displayed a faster and more pronounced cognitive decline in comparison to the parental lines. Due to its stronger phenotype the novel TAPS line has qualified itself as a useful new tool to study AD pathophysiology, and for preclinical studies testing new therapeutic options.

## 2. Results

### 2.1. TAPS Mice Accumulate Aβ Aggregates in the Striatum, Hippocampus, and Cortex as Early as 6 Months

TAPS mice were viable and fertile but showed a 14% increased rate of premature death in comparison to wild-type (WT) littermates. For comparison, APP/PS1 mice showed a 3% increased rate of premature death (Appendix A). Both, TAPS and APP/PS1 mice, developed an increasing amyloid pathology with neuritic plaques in the brain over time. With an earlier onset, at the age of 6 months, TAPS mice showed plaque formation starting in the cortex, hippocampus, and also lateral striatum. Over time, all mentioned regions underwent a constant increase in plaque density, with the highest amounts in the cortex and slightly less Aβ plaques in the hippocampus. In APP/PS1 mice, visibly less plaque formation could be found at the same age in the cortex and emerged to the hippocampus with 9 months but with nearly no Aβ accumulation in the striatum. Overall, plaque formation in early ages was visibly lower than in the corresponding TAPS mice, and increasing in the cortex and hippocampus to a comparable level in later life. The cerebellum showed only little Aβ accumulation over time and accumulation in the thalamus could be observed in both genotypes. In contrast to the previously mentioned mouse lines, the TBA2.1 mice developed decent amounts of Aβ aggregates in the striatum, already at the age of 6 months and kept those levels until older ages. However, there was nearly no Aβ accumulation visible in brain regions other than the striatum.

To investigate the composition of the Aβ plaques in all mouse lines, a double staining was accomplished with antibodies against truncated pyroglutamate Aβ at position 3 (pE3Aβ) and total Aβ (antibody 6E10) in 24-month-old mice. TAPS and APP/PS1 mice showed an intense staining of plaques in the cerebral cortex for both, Aβ and pE3Aβ, as shown in Figure 1, and comparable results were found also in the hippocampus. It could be seen that compact neuritic plaques, as well as diffuse Aβ, were positively stained with 6E10 in both cases. Albeit the 6E10 signal was stronger, in diffuse Aβ, a minor portion of pE3Aβ could be observed as well, indicating a possibly lower content of those truncated Aβ species than full-length Aβ in diffuse accumulations. In compact plaques, however, pE3Aβ was more prominent in the center core of the plaques than in the surrounding. In principle, the overall plaque morphology and distribution of pE3Aβ in the cortex and hippocampus was comparable between TAPS and APP/PS1 mice. TBA2.1 mice, however, showed no visible Aβ accumulation in the cortex, as well as in the hippocampus, and were not distinguishable from WT mice in those regions. Differences, however, could be seen in the striatum (caudate putamen) of the mouse lines, which are shown in Figure 2.

TAPS mice showed a recognizable accumulation of Aβ in the lateral striatum (Figure 2). The signal was almost solely congruent between total Aβ (antibody 6E10) and pE3Aβ (antibody pE3). The TBA2.1 mice accumulated aggregates in the striatum that were positive for pE3Aβ and 6E10, although Aβ aggregates were much smaller compared to TAPS. Delineation of those aggregates in TBA2.1 was also sharper than in TAPS, possibly indicating an intracellular accumulation in this line. APP/PS1 mice, however, showed nearly no accumulation of Aβ species in the corresponding striatal region, compared to TAPS littermates. In summary, TAPS and TBA2.1 mice showed Aβ accumulation in the lateral striatum, in contrast to APP/PS1 mice.

Looking closer at the cellular response of the brain, a staining against glial fibrillary acidic protein (GFAP) was accomplished. Reactive astrogliosis could be demonstrated in investigated brains between 6 and 18 months. Exemplarily showing the results, comparing 18-month-old brains in Figure 3, the intense staining of Aβ with 6E10 (red) can be shown for the cerebral cortices of TAPS and APP/PS1 mice, as previously mentioned. Acute astrogliosis surrounded the direct vicinity to amyloid plaques, as visible in the cortex images (Figure 3), but Aβ presence also seems to promote general activation of astrocytes throughout the whole brain in TAPS and APP/PS1 mice. Although there was nearly no Aβ in the striatum of the latter, a certain reactive astrogliosis could be observed in this area as well. Overall, this indicates a strong increase in neuroinflammatory processes in both mouse lines and a general activation of astrocytes throughout the brain, also in areas without plaque pathology, e.g., the striatum. TBA2.1 mice also showed a certain degree of astrogliosis for the regions of the lateral striatum, since there were also Aβ aggregates visible. Though, the astrocytes in the rest of the brain were not distinguishable from the WT. Even in WT mice, some GFAP-positive astrocytes could be found, mostly in white matter. The molecular layer of the cerebellum was roughly free from reactive astrocytes in all investigated brains, in concordance with poor Aβ accumulation in this area.

Since neuron loss in the CA1 region of the hippocampus was described for homozygous TBA2.1 mice [28], we analyzed this brain region also in the TAPS line. Quantification of neuronal nuclei in the hippocampus of 24-month-old TAPS and WT mice showed no significant differences in neuronal density in the stratum pyramidale of the CA1. Cell counts were on comparable levels; therefore, no detectable signs of neurodegeneration could be observed in the designated area for TAPS compared to WT mice (Appendix A).

To investigate differences in Aβ load between the mouse lines in more detail, a quantification with ImageJ was accomplished. The number of neuritic plaques and the average size were quantified in the areas of the cortex, striatum, and hippocampus, for TAPS and APP/PS1 (Figure 4). TAPS mice displayed in general a higher number of plaques than APP/PS1 mice, which was significant in all brain regions analyzed (two-way ANOVA; cortex, genotype *p* = 0.007, age *p* < 0.001; hippocampus, genotype *p* < 0.001, age *p* < 0.001; striatum, genotype *p* < 0.001, age *p* < 0.001, genotype x age *p* < 0.001). At the age of 7 months, TAPS mice already had a significantly higher level of Aβ in the cortex, compared to their APP/PS1 counterparts (multiple *t*-test; *p* = 0.03). They built approximately five times more deposits at the same age in the cortex and showed a detectable amount of neuritic plaques in the striatum, as well. At the age of 15 months, both, TAPS and APP/PS1 mice, showed a comparably high plaque load in the cortex but with a higher increase in TAPS after 18 months. The increase in plaque formation of APP/PS1 slowed remarkably down after 15 months in the cortex and hippocampus, trending towards a plateau at this time.

Hippocampal plaque formation could also be observed at a 10 times higher level in 7-month-old TAPS in comparison to APP/PS1 (multiple *t*-test; *p* = 0.009). At the age of 15 months, both, TAPS and APP/PS1 mice, showed a comparably high plaque load in the hippocampus. With 18 months, TAPS mice, however, displayed a significant increase in hippocampal plaque load (multiple *t*-test; *p* = 0.003), whereas numbers in APP/PS1 mice remained on a relatively constant level.

In the striatum, only TAPS mice developed an amyloid pathology with neuritic Aβ plaques. Differences against APP/PS1 could be demonstrated to be significant at 7 (multiple *t*-test; *p* = 0.006), 15 (multiple *t*-test; *p* = 0.009), and 18 months (multiple *t*-test; *p* < 0.000001). APP/PS1 mice showed only constantly low levels of neuritic plaques in the striatum, even in older individuals.

Analyzing the plaque size at 15 and 18 months of age, it could be shown that TAPS mice tend to have slightly smaller plaques in the cortex (879.3 ± 26.9 µm², 15 months; 915.7 ± 52.5 µm², 18 months) and hippocampus (994.63 ± 54.0 µm², 15 months; 992.8 ± 110.2 µm² 18 months), compared to the APP/PS1 cortex (922.2 ± 22.6 µm², 15 months; 1080.3 ± 21.5 µm², 18 months) and hippocampus (1111.1 ± 53.1 µm², 15 months; 1083.0 ± 50.1 µm², 18 months). Both genotypes showed a constant size of plaques in both areas over the analyzed time. The aggregates in the striatum of TAPS mice, however, were significantly smaller in size, with 588.8 ± 34.3 µm² for 15 months, and 586.5 ± 52.4 µm² for 18 month-old-mice (two-way ANOVA brain region, *p* < 0.001; Holm-Sidak´s post-hoc, cortex vs. striatum *p* = 0.001, hippocampus vs. striatum *p* = 0.001), compared to those in the cortex and hippocampus.

### 2.2. TAPS Mice Show Phenotypic Alterations in the SHIRPA and Open Field Tests

In the SmithKline Beecham Pharmaceuticals; Harwell, MRC Mouse Genome Centre and Mammalian Genetics Unit; Imperial College School of Medicine at St Mary’s; Royal London Hospital, St Bartholomew’s and the Royal London School of Medicine; Phenotype Assessment (SHIRPA) test, TAPS mice were initially compared to their WT littermates for development of a behavioral phenotype with increasing age. TAPS mice showed a consistent increase in scores throughout the whole examination time, as it can be seen in Figure 5. At 4 and 6 months of age, changes were relatively small compared to older animals. At 9 months of age, the average scores increased to 1.6 and further to a score of 3.9 with 12 months. The latter was highly significant compared to the values of the WT littermates (mixed effects analysis, genotype *p* < 0.0001, age *p* < 0.0001, interaction *p* < 0.0001; Sidak’s post-hoc test 12 m, *p* = 0.0001). Over time, scores raised to 4.9 at 15 months (Sidak’s post-hoc test 15 m, *p* = 0.0002) and finally 6.6 at 18 months of age (Sidak’s post-hoc test 18 m, *p* = 0.0273), which both proved to be statistically significant compared to WT. Due to the partly cross-sectional type of this study, not all mice from the initial group endured the full investigation period of 18 months. Male and female animals were pooled for analysis. The most common hallmark of the TAPS phenotype in the SHIRPA test was a reduction in sensory perception, mostly prominent in the pinna reflex, startle response, and the flank pressure. A larger number of animals also showed deficits in the hanging behavior.

Summarizing the results of the WT, the scores remained on a relatively low level throughout the whole examination period. The scores varied on average around 0 and 0.4 for all age groups. Single mice showed minor abnormalities, which were transient and did not exceed a score of one. The most common observation was a slight reduction in movement in the cage and a reduction in the time at the hanging behavior, which could be correlated with a high body weight. Analysis of the bodyweight over time demonstrated that overall, there was no genotype-dependent discrepancy observable for both tested genders (Appendix A).

A second cohort of TAPS mice was further tested at the age of 18 months together with their littermates in several behavioral tests. In the open field test, which was done once at 18 months, APP/PS1 and TBA2.1 did not differ significantly from the WT mice. However, TAPS mice seemed to be faster (Figure 6A), to travel more (Figure 6B) and to be more active (Figure 6C) than the WT, indicating that they might display hyperactive behavior. They spent the same amount of time in the center and the border of the open field (Figure 6D), which indicates that none of the mice had reduced exploration or increased anxiety. Taking these results together, TAPS mice might have alterations in the general spontaneous activity in the open field test.

### 2.3. TAPS Mice Develop Cognitive Deficits in Different Behavioral Tests

Several cognitive behavioral tests were conducted in order to characterize the cognitive abilities of TAPS mice in comparison to their littermates. In the novel object recognition test (NOR), only the WT mice explored significantly more the novel than the familiar object (paired *t*-test, *p* = 0.0380), indicating functional memory for the familiar object. For all other lines, exploration times between the novel and familiar object did not reach statistical significance, although there is some trend for more exploration of the novel object (Figure 7). High variability and low animal numbers, and low overall exploration may be accounted for this in case of the TAPS and APP/PS1 line, respectively. Regarding total exploration time of the objects, TAPS mice showed a tendency towards higher exploration of both objects; however, this did not reach statistical significance.

The T-maze spontaneous alternation was measured to assess short-term memory. With 18 months of age, all groups had similar amounts of alternations (Appendix A). However, with 20 months of age, APP/PS1 (one-way ANOVA, *p* = 0.0192; Holm-Sidak’s post-hoc test, *p* = 0.0349), TAPS (Holm-Sidak’s post-hoc test, *p* = 0.0349) and TBA2.1 (Holm-Sidak’s post-hoc test, *p* = 0.0145) mice alternated less than the WT (Figure 8). These results indicated that they were not able to discriminate which arm was visited previously. In comparison, the WT mice were able to discriminate the previous arm, which indicates an intact working memory. Moreover, only WT mice alternated significantly above chance (i.e., >50%) (one sample *t*-test, *p* = 0.0001). Therefore, they were able to choose the new arm instead of entering an arm randomly.

With 18 months of age, the TAPS mice froze less than the WT in the cued fear conditioning paradigm (one-way ANOVA, *p* = 0.0015; Dunnett’s post-hoc test, *p* = 0.0042) (Figure 9A). This result indicated that they were not able to associate the sound (cue) with the shock in the habituation phase, implying a deficit in the associative learning process. On the other hand, all groups froze the same amount in the contextual fear conditioning part of the test, meaning they were all able to associate the arena (context) with the shock (Figure 9B). With 20 months of age, the test was repeated with the same cohort of mice. This time the TAPS mice froze less than the WT mice also in the contextual fear conditioning (Appendix A), indicating impairment in the contextual memory at older ages.

With 20 months of age, the Morris water maze test (MWM) was carried out in order to measure the spatial learning and memory abilities of the mice. During the training session, TAPS mice took longer to find the platform compared to the WT (Figure 10A, mixed effects analysis, genotype *p* = 0.0210, days *p* < 0.0001, interaction *p* = 0.5794; Dunnett´s post-hoc test, *p* = 0.0087). In contrast, APP/PS1 mice were able to find the platform faster along the days of training. Additionally, TBA2.1 mice took longer to find the platform compared to WT only on the fourth day, showing a delay in learning (Dunnett´s post-hoc test, *p* = 0.0241). In conclusion, TAPS mice displayed a learning deficit since they were not able to memorize the platform location in the pool. In the probe trial, both TAPS and APP/PS1 mice had a tendency towards reduced searching time in the target quadrant compared to WT, but this difference did not reach statistical significance, presumably due to insufficient animal numbers. Interestingly, TBA2.1 mice also had reduced time searching in the target quadrant compared to WT (Figure 10B, one-way ANOVA, *p* = 0.0357; Dunnett’s post-hoc test, *p* = 0.0319).

## 3. Discussion

In recent years, a number of drug candidates have been tested in clinical trials to find a new therapeutic option against AD, but almost all failed due to lack of efficacy. Many of the new substances had previously demonstrated efficacy in transgenic mouse models, which is why the mouse models have come under criticism. Although transgenic mouse models have proven to be a valuable tool for studying the pathophysiology of AD, they are incomplete models of the human disease. Most transgenic mouse lines are able to mimic only a few aspects of the disease, i.e., neuritic amyloid plaques, tauopathy, neuroinflammation, cognitive decline, or neurodegeneration [30].

In the current study, we describe the generation of a novel transgenic mouse line as an attempt to generate an improved model harboring a combination of AD-relevant hallmarks, especially the aggressive phenotype produced by pEAβ, an abundant formation of neuritic plaques, and extensive cognitive decline. Thus, the TAPS mice were developed in order to understand the role of pEAβ and its interaction with other Aβ species that are constitute of neuritic plaques. We characterized the progression of cerebral amyloidosis, and the development of general and cognitive behavioral deficits in this novel triple transgenic mouse line.

As expected, TAPS mice developed a more severe phenotype in comparison to its heterozygous parental lines. Like APP/PS1 mice, they developed neuritic plaques throughout the brain, especially in the cortex, thalamus, and hippocampus, and little pathology in the cerebellum. Most striking, and in addition to what can be observed in the APP/PS1 line, TAPS mice also showed Aβ aggregates in the lateral striatum, which appeared larger and more mature than the small Aβ particles present in the striatum of the parental TBA2.1 line. We could confirm data by Alexandru et al. [28] that heterozygous TBA2.1 mice at 21 months had only a low amount of N-terminally truncated and pyroglutamate-modified Aβ in small intracellular aggregates in the striatum, which were, however, not sufficient to induce motor deficits in this line. The clinical relevance of Aβ aggregates in the striatum are not yet clear, although striatal amyloid plaques have also been found in AD patients. Striatal Aβ depositions have been described mainly in AD patients of advanced stages [31,32,33,34] but may also occur in the preclinical stage [35]. A recent study showed that Aβ deposition in the striatum correlates with both, memory deficits and tau pathology [36]. Nearly all amyloid plaques found in TAPS mice stained positive for both total Aβ and pE3Aβ. PE3Aβ was mainly located in the plaque core of compact and neuritic plaques and less in diffuse Aβ deposits. The 6E10 antibody, used in this study to detect total Aβ, has the capability to bind both, full-length Aβ as well as N-terminally truncated species, while the pE3Aβ antibody exclusively binds to truncated pyroglutamate Aβ. Apart from the striatum, neuritic plaques in other brain regions of TAPS mice were also composed of pE3Aβ in addition to other Aβ species. Thus, the TAPS mice had an amyloidosis phenotype combining the distribution patterns from both the APP/PS1 and TBA2.1 mice, but with earlier onset and faster progression. We characterized TAPS mice with a significantly earlier onset of plaque formation in the cortex at the age of 6 to 7 months, whereas APP/PS1 reached comparable Aβ levels later in life. Additionally, deposits in the hippocampus formed at a younger age than in APP/PS1. TAPS showed a constant increase in plaque counts over time, whereas in APP/PS1, progression seemed to slow down towards a plateau after 15 months of age. Taken together, the additional production of the aggregation-prone pE3Aβ species led to both, an accelerated plaque deposition and higher amyloid load at older ages. With regard to plaque size, the TAPS mice developed slightly smaller but more plaques than the APP/PS1 mice at the same age. In a previous study, Frost et al. demonstrated the development of pEAβ-positive plaques in APP/PS1 mice after full-length Aβ deposits formed [24]. This could also be shown in the present study, because, apart from plaque size, old APP/PS1 mice displayed a similar plaque composition than TAPS animals in the cortex and hippocampus. The smaller plaque size of TAPS mice in comparison to the parental APP/PS1 line seemed to be the consequence of early pE3Aβ presence in the brain, possibly facilitating the formation of fibrils and generating smaller plaques through faster aggregation. The specific location of higher pEAβ levels in the center core of compact plaques has also been observed in tissue from human patients and underlines the theory of pEAβ as a seeding spot for other Aβ species or diffuse Aβ in brains [37,38].

Additionally to plaque formation, both APP/PS1 and TAPS mice developed progressive neuroinflammation in the brain, indicated by high numbers of reactive astrocytes around plaques. Neuroinflammation is one of the hallmarks of AD, which contributes to cognitive deficits in human and animals [39]. Reactive astrocytes and microglia exacerbate the Aβ toxicity and neuronal death. Heterozygous TBA2.1 mice, as expected, did not display an overall increase in astrogliosis since they do not develop plaques. Hence, a certain degree of activation could be seen in the striatum, in the vicinity of small Aβ particles. Regarding the neuroinflammatory response, the TAPS mice seemed to develop even higher levels of astrogliosis, compared to APP/PS1 mice. Taken together, the higher activation of astrocytes in the striatum and in other brain regions could have contributed, in conjunction with early production of the highly toxic pE3Aβ species, to more severe cognitive deficits in TAPS mice, compared to the parental lines.

Despite showing a strong reactive astrogliosis and an increased inflammatory milieu in the investigated brain areas, together with a high plaque load, however, neurodegeneration could not be observed in the hippocampus of TAPS mice, compared to WT. The CA1 region of the hippocampus, in particular the stratum pyramidale, showed comparable amounts of neurons throughout the investigated area of TAPS and WT mice. Although neurodegeneration in this brain region was described for homozygous TBA2.1 mice [28], obviously, the heterozygous status of the TBA2.1 transgene was not sufficient to induce neuron loss in TAPS mice. However, this does not exclude the presence of subtle neurodegenerative changes throughout the entire brain, which we might have overlooked.

TAPS mice developed cognitive deficits beginning with 18 months of age mainly in the cued fear conditioning. Later, at 20 months of age, they also displayed a deficit in T-maze and MWM. Those different behavioral tests evaluated different types of cognitive abilities, which are processed by different areas of the brain. The MWM measures spatial memory [40]. The spatial memory processing occurs primarily in the hippocampus; therefore, lesions in this brain area have been shown to induce cognitive deficits [41,42]. It has been shown before that amyloid pathology in the hippocampus, consisting of plaques and, more importantly, soluble toxic Aβ oligomers, can induce synaptic loss and cognitive deficits [27,43,44,45], which could also explain the spatial memory deficits in TAPS mice. As seen in histological investigation, TAPS mice showed, in contrast to APP/PS1 mice, a constant increase in hippocampal plaque formation, even at higher ages, possibly facilitating this effect. The spontaneous alternation paradigm in the T-maze is a measure of the spatial working memory, and the NOR is based on working memory and recognition memory, too, which are processed in both the hippocampus and cortex [46,47,48,49,50]. Since Aβ pathology is also abundantly present in the cortex of TAPS mice, deficits in working memory can be explained by toxic Aβ species in the cortex as well. Finally, the cued fear conditioning paradigm is based on fear memory with involvement of the amygdala [49,50], a brain region that was also affected by Aβ pathology in TAPS mice. Deficits in fear memory, particularly in the cued fear conditioning, have also been observed in AD patients [51,52] and other transgenic AD models before [53,54].

Even though TBA2.1 mice do not develop Aβ plaques, they showed clear deficits in both the T-maze and MWM. Cognitive deficits have not been reported for heterozygous mice of this line before, and support the importance of pEAβ for neurotoxicity in the absence of amyloid plaques. A neurotoxic pEAβ-dependent process seemed to be responsible for the working and spatial memory deficits in those mice. Moreover, the APP/PS1 mice developed abundant Aβ plaques in the same brain areas as TAPS mice, without deficits in MWM and fear conditioning tests, unlike TAPS mice at the same age. Therefore, the earlier aggregation of Aβ, most probably induced by pEAβ, and pEAβ´s known neurotoxic potential [55,56], could have accelerated the cognitive decline in TAPS mice.

Besides their cognitive deficits, TAPS mice also developed sensory and motor impairments. In the SHIRPA test, TAPS mice showed phenotypic alterations beginning at 12 months of age compared to both WT and younger TAPS mice. As shown before, TBA2.1 mice developed similar alterations starting at 21 months of age [29]. Therefore, it can be concluded that the combination of pEAβ and full-length Aβ in the brain might be responsible for acceleration of the phenotype in comparison to pEAβ alone. In addition, we observed a trend towards higher velocity, longer distance travelled, and generally more activity in the open field, indicating a hyperactive phenotype of TAPS mice compared to WT and TBA21 mice. One might speculate whether these motoric alterations could be due to the development of Aβ pathology and neuroinflammation in the striatum. The striatum, as a key interface for excitatory and inhibitory neurons, plays a major role in action selection and motor function [57]. Moreover, in other AD mouse models, the observed hyperactivity was correlated with basal ganglia circuitry dysfunction [58,59,60]; therefore, one might assume that striatal Aβ deposits induce changes in the basal ganglia network. So far, the relevance of a hyperactive phenotype in AD mouse models is not quite clear. Although hyperactivity is also part of the behavioral and psychological symptoms of dementia in patients, its causes are still under discussion and have not been sufficiently investigated yet (for a review, see [61]). Additionally, TAPS mice showed a reduced sensory response in the SHIRPA test. Balsters et al. demonstrated the connectivity of the lateral striatum (caudate putamen) to cortical areas in the motor-cortex (m1) and somatosensory cortex (s1 and s2) in mice [62]. A disturbance in signal transduction in this area could promote the deficits TAPS mice showed in contrast to their parental lines at the same age. It leads to the assumption that the combination of deposits in the striatum with those in the cortex, thalamus, and hippocampus and the elevated response of astroglia in this area could lead to the hyperactivity pattern and sensory deficits seen in these mice. Some studies also demonstrated a hyperactive behavior of APP/PS1 mice in the open field at younger ages (4 to 8 months) [63,64], which was explained by a decrease of endocannabinoids in the striatum [64]. A similar process can be assumed for TAPS mice, but more studies are needed to prove this connection.

Unlike previous reports, the APP/PS1 mice used in the current study did not develop cognitive deficits in the MWM even at an age of 21 months. One reason for that discrepancy might be the variety of used protocols and differences in the housing conditions [65,66]. However, a non-significant trend towards a deficit could be observed in MWM. This could be due to the relatively small groups of mice included in some of the behavioral tests, whose variability prevented a slight deficit from becoming statistically significant. Concerning heterozygous TBA2.1 mice, little is known about their cognitive abilities, so far. We have described before, that, in contrast to the homozygous animals, the heterozygous TBA2.1 mice did not develop any motor deficits, but a higher score in the SHIRPA test at 21 months of age [29]. Corroborating with this previous study, the TBA2.1 parental line did not show any conspicuities, including activity, exploratory, or anxiety-related behavior, in the open field test at 18 months.

Finally, the new TAPS line is an amyloidosis model that reflects several aspects of the human disease. However, the aspect of tauopathy, which is an important feature of human AD, is missing. Since none of the previously described amyloidosis models show excessive tau phosphorylation and also no neurofibrillary tangles [67], development of tauopathy in TAPS mice was not to be expected. In general, the behavioral deficits of all three mouse lines, and especially the APP/PS1 and TAPS mice, appeared with an advanced age of 18 to 20 months, which might limit their practical use for future studies. On the other hand, mouse models with very aggressive phenotype progression and early behavioral deficits before clear pathological alterations have been criticized because their relevance to the clinical disease has been questioned [67]. In this respect, the new TAPS line joins the ranks of the rather late AD models in which the cognitive deficits clearly develop as a consequence of pathological processes.

In conclusion, we were able to demonstrate an accelerated amyloid pathology in TAPS mice with earlier onset and increased Aβ deposition induced by pE3Aβ. In addition, the TAPS mice developed faster and more pronounced cognitive deficits than the parental lines as measured by several cognitive behavioral tests. Both parental mouse lines, TBA2.1 and APP/PS1, have been successfully used for preclinical therapeutic studies in which new substances were tested during their development as drug candidates against AD [68,69]. The novel TAPS mouse line combines their advantages by increasing the neurotoxic pE3Aβ species and inducing robust cognitive deficits, and thus, qualifies itself as a useful new amyloidosis model for future preclinical studies testing new therapeutic options.

## 4. Materials and Methods

### 4.1. Animals

TBA2.1 mice were a generous gift from Probiodrug AG (Halle, Germany) and were bred in house by mating of heterozygous mice. The mice were originally described on a C57BL/6 × DBA1 background and were further crossed to a C57BL/6 background for more than four generations. As described by Alexandru et al. [28], the transgene of the TBA2.1 line was designed for chromosomal integration by addition of cDNA sequences of a pre-pro-peptide of murine thyrotropin releasing hormone (TRH) fused with the modified human Aβ polypeptide Aβ(Q3–42) under a neuron-specific promotor. Aβ(Q3–42) is expressed in neuronal cells and subjected to the secretory pathway, where it is post-translationally modified by the endogenous glutaminyl cyclase into pEAβ3-42. Homozygous TBA2.1 mice develop a motor-neurodegenerative phenotype as a consequence of pEAβ3-42-induced neurotoxicity starting with 2 months and progressing further until the age of 5 months, when the humane endpoint is reached. In addition, an age-related massive neurodegeneration can be observed, especially in the hippocampus, and deposition of small Aβ aggregates in brain regions, such as the hippocampus and striatum, accompanied by neuroinflammation. Heterozygous TBA2.1 mice display a milder phenotype starting with 21 months of age [29].

APPswe/PS1ΔE9 mice were introduced by Jankowsky et al. [8] and express both a chimeric mouse and human Amyloid Precursor Protein (APP695swe) and human presenilin 1 mutated by a deletion of exon 9 (PS1ΔE9) [11] under the control of neuron-specific promotor elements. The mice develop neuritic Aβ plaques and neuroinflammation beginning with 6 months, and cognitive deficits, which are detectable in the Morris water maze test [17]. APPswe/PS1ΔE9 mice on a C57Bl/6 background were received from the Jackson Laboratory (Bar Harbor, MA, USA) and bred in-house by mating of heterozygous and wild-type mice of this line.

T(TBA2.1)APS(APPswe/PS1ΔE9) mice were generated by crossing heterozygous TBA2.1 and heterozygous APPswe/PS1ΔE9 (APP/PS1) mice in the C57BL/6 background. The resulting TAPS mice were heterozygous for both TBA2.1 and APP/PS1 transgenes. For behavioral tests and histological examinations, male and female littermates of these matings were used with the following genotypes: wild-type (WT), APP/PS1, TBA2.1, and TAPS (Table 1). All mice were heterozygous for the respective transgene.

All mice were bred in-house with a 12/12 h light/dark cycle, constant temperature of 22 °C, and 54% humidity. Food and water were available ad libitum. All behavioral experiments were performed in accordance with the German Law on the protection of animals (TierSchG §§ 7–9) and were approved by the local ethics committee before the start of the experiments (Landesamt für Natur, Umwelt und Verbraucherschutz, North Rhine-Westphalia, Germany, numbers 84-02.04.2011.A359, 84-02.04.2014.A362, 81-02.04.2018.A400, 81-02.04.2019.A304 were approved on 09 December 2014, 05 February 2019, 21 February 2019 and 21 January 2019, respectively).

Based on the late behavioral alterations the parental lines displayed in our hands in the past, the cognitive test battery was started at 18 months of age. The first cohort of TAPS and WT littermates was tested in the SHIRPA test battery for initial check of behavioral abnormalities. Then, two more cohorts of 18-month-old littermates were tested repeatedly (at 18 and 20 months of age) in the following behavioral tests to test their cognitive abilities: cohort 2, T-maze (18 and 20 months), MWM (20 months); cohort 3, open field (18 months), novel object recognition (NOR, 18 months), T-maze (18 and 20 months), cued and contextual fear conditioning (18 months), contextual fear conditioning (20 months), and MWM (20 months). Three TAPS, two APP/PS1 mice, one TBA2.1, and one WT mouse died during the longitudinal testing between the age of 18 and 20 months. Three APP/PS1, two TBA2.1, and two WT mice had to be excluded from the T-maze due to inactivity.

### 4.2. Histology

For histological studies, mice from all genotypes were used at different ages from 6 to 24 months (Table 1). Mice were killed by cervical dislocation, and brains were taken and frozen at −80°C until further processing. Right brain hemispheres were used to generate 20 µm sagittal sections with a cryostat (LEICA Biosystems, Wetzlar, Germany). Immunofluorescence staining was performed in order to evaluate the Aβ-plaque/particle distribution and size with mouse monoclonal antibody anti-Aβ, Clone 6E10 recognizing the N-terminal Aβ strain (1:200; BioLegend, San Diego, CA, USA), as well as neuroinflammation (activated astrocytes) with polyclonal rabbit anti GFAP antibody (1:1000; Dako Omnis, Agilent, Santa Clara, CA, USA). Plaque morphology and composition was analyzed with brains from 24-month-old mice using a double-staining against pE3Aβ with the rabbit anti-Abeta-pE3 antibody (1:500; Synaptic Systems GmbH, Goettingen, Germany) in combination with the 6E10 antibody against whole Aβ.

Briefly, frozen sagittal sections of the brains were fixed in 4% buffered formaldehyde solution. Afterwards, the slides were incubated in 70% formic acid as antigen retrieval and blocked with M.O.M (Mouse on Mouse) blocking reagent (Vector Laboratories Inc., Burlingame, CA, USA) to prevent unspecific binding of the primary antibody. The slides were then incubated overnight with the primary antibody solution in 1% TBS buffer and containing 1% BSA at 4 °C in a humid chamber. The next day, slices were washed in buffer and incubated with the secondary antibodies, diluted in the same solution as the primary ones for 2.5 h at RT: Goat anti-Mouse IgG (H+L) Alexa 568 and Goat anti-Rabbit IgG (H+L) Alexa 488 (1:300; Thermo Fisher Scientific, Waltham, MA, USA). For assessment of the cell nuclei, a DAPI staining (5 µg/mL, Sigma Aldrich, Steinheim, Germany) was performed after washing the slides consecutively to the secondary antibody incubation. Subsequently, slices were mounted with Aqua Poly Mount (Polysciences, Inc., Warrington, Pennsylvania, USA), coverslips and stored at 4 °C until further analysis at the microscopes.

For slice evaluation, overview images were made with a Zeiss Lumar V12 SteREO (Carl Zeiss AG, Oberkochen, Germany) at the corresponding fluorescent channels for the antibodies. In total, 16 animals were used for TAPS (7 months *n* = 7; 15 months *n* = 5; 18 months *n* = 4) and 15 animals for APP/PS1 quantification (7 months *n* = 8; 15 months *n* = 3; 18 months *n* = 4). Nine images per animal were obtained for each channel at a 10× magnification. Those images were used for quantification of plaques with the image analyzer ImageJ v1.48.

For analysis of the plaque pattern and the reactive astrogliosis on a smaller scale, images were obtained by the Leica LMD 6000 Fluorescent Microscope (Leica Microsystems GmbH, Wetzlar, Germany) at a magnification between 100× and 400×. Overlay images were created via the company’s software Leica Application Suite v4.5 (LAS). Images obtained with the LMD were used for qualitative measures and for possible discrimination of differences between the investigated genotypes.

For analysis of neuron loss, a subset of six slides per animal was analyzed from 24-month-old TAPS (*n* = 5) and WT mice (*n* = 5). After fixation in 4% buffered formalin, slides were stained for 5 min in 5 µg/mL DAPI solution (4′,6-Diamidin-2-phenylindol, Sigma Aldrich, Steinheim, Germany). Microscope images of the hippocampus were taken at 50× magnification with the Leica LMD 6000 Fluorescent Microscope. The CA1 region of the hippocampus was evaluated for the number of positive nuclei via Cell Profiler Software (version 2.0.10415, Broad Institute, Cambridge, USA).

### 4.3. Behavioral Tests

#### 4.3.1. SHIRPA

The SHIRPA test battery was adapted from [70]. This test evaluates the general phenotypic alterations in transgenic mice, especially on their sensoric and locomotoric capacities and has been demonstrated to be very sensitive towards the detection of deficits in different transgenic mouse lines, including the TBA2.1 model [29,71]. The number of mice used in this test are given in Table 1. The test was divided into two parts: (1) Observation in an empty cage and (2) handling. In each part, several criteria were scored compared to WT littermates from 0 (no difference) to 3 (strong difference). For the analysis, each criteria score was summed to a final score per animal. The examined tasks in the SHIRPA consisted of the examination in the cage for abnormalities in posture, gait, alertness, frightening, pain reaction, and the Straub-tail test. For the examination on the hand, the animals were tested for reaction towards handling (gentle pressure on the flank of the animal), pinna reflex, forelimb placing reflex, and the hanging behavior on a thin pole. Additionally, the body weight of the animals was recorded.

#### 4.3.2. Open Field Test

The open field test is used to evaluate the general spontaneous behavior in rodents as well as anxiety. In a cubicle arena (40 cm × 40 cm × 40 cm), mice (for exact numbers, see Table 1) were allowed to explore the arena freely for 30 min. The arena was imaginarily divided into 12 squares to determine the border, center, and corner zones. Mice were observed with a camera driven tracking system (EthoVision XT 15.0.1416, Noldus Information Technology, Wageningen, The Netherlands). The following parameters were evaluated: active time, velocity, distance travelled, time spent in the center, in the border, and in the corner.

#### 4.3.3. Novel Object Recognition Test

In the same arena where the open field was performed, one day later, the mice (Table 1) explored two similar objects (familiar) for 10 min. After 20 min of memory retention interval, mice were placed again in the arena. One familiar object was replaced by a new one (novel) and they were allowed to explore for another 10 min. Exploration time for an object was considered when mouse placed the nose at least within 2 cm distance into the direction of the object. Mice were observed with a camera driven tracking system (EthoVision XT 15.0.1416, Noldus Information Technology, Wageningen, The Netherlands). For the analysis purpose, the time mice spent exploring each object was taken.

#### 4.3.4. T-maze Spontaneous Alternation

The protocol was adapted from Spowart-Manning and Van der Staay [72]. The maze consisted of three arms (31 cm × 10 cm each): the left and right arm can be closed by a gate and the start arm (perpendicular to the left and right) has no gate. In the first trial, mice were forced to enter one goal arm by closing the door of the other arm. Once the mice returned to the start arm, both arms could be freely accessed. After the mice entered one arm, the other one was closed by a door, and it was counted as one trial. This was repeated for a maximum of 14 trials or 15 min. If a mouse did not reach seven trials, it was excluded from the analysis. For analysis, the spontaneous alternation was calculated by the number of correct alternating choices divided by the total amount of trials. For the number of mice included in this test, please refer to Table 1.

#### 4.3.5. Contextual and Cued Fear Conditioning

The protocol for the fear conditioning was adapted from Curzon et al. [73]. First, the mice were placed in the chamber (17 × 17 × 25 cm; Ugo Basile, Gemonio VA, Italy) for 120 s of habituation. Then, a combination of conditioned stimuli [5] and unconditioned stimuli (US) were presented for three times with a 60 s interval. A 3 s CS tone (50%; 2000 Hz) followed, and a US foot shock was given during the last 2 s of the CS (0.35 mA). Before the mice were returned to the cage, they stayed in the arena for an additional 60 s. Then, 24 h later, the mice were placed in the same testing chambers used on day one for 5 min (contextual). After 25 min, the mice were placed in a new environment that could be explored freely for 180 s. Then, the CS tone was played three times for 30 s at 60 s intervals, similar to the habituation phase (cued). All mice (Table 1) were observed with a camera-driven tracking system (EthoVision XT 15.0.1416, Noldus Information Technology, Wageningen, The Netherlands). The following parameters were analyzed in each session: the percentage of freezing behavior as detected by the software.

#### 4.3.6. Morris Water Maze

The protocol for the MWM was modified after Morris [74]. In brief, mice were placed in a circular pool (diameter of 120 cm × 60 cm height) with a hidden platform (diameter of 10 cm × 31. 5 cm height) in a fixed position. In the training, four trials per mouse per day (5 days in total) were performed. The maximum time of each trial was 1 min and the mice started in each trial in a different position. One day after the last training day, the probe trial was performed, where the platform was removed, and mice had to swim for 1 min. All mice (Table 1) were observed with a camera-driven tracking system (EthoVision XT 15.0.1416, Noldus Information Technology, Wageningen, The Netherlands). The time and distance needed for finding the platform in each training section and the time they explored the target quadrant in the probe trial were analyzed.

### 4.4. Statistics

For the statistical analysis of behavioral tests, GraphPad Prism v8 (GraphPad Software, San Diego, CA, USA) was used. The normality of the data was checked by visualization of the Normal QQ plot. The SHIRPA test was analyzed by mixed effect analysis with Tukey’s and Sidak’s post-hoc test, to compare age and genotype, respectively. The different parameters of the open field test were analyzed by one-way ANOVA and Dunnett’s post-hoc test. The NOR was analyzed with the paired t-test, where the exploration time of the novel object was compared to the exploration time of the familiar object for each mouse. The analysis of the T-maze was calculated via one-way ANOVA and Holm-Sidak’s post-hoc test, to be compared to WT mice. One sample t-test was accomplished against the theoretical mean of 50%. For both the cued and the contextual fear conditioning test, a one-way ANOVA and Dunnett’s post-hoc test was the chosen statistical analysis. In the MWM training analysis, mixed effect analysis and Dunnett’s post-hoc were used. To evaluate the MWM probe trial, the one-way ANOVA and Dunnett’s post-hoc test were used. The analysis of plaque quantification was performed with SigmaPlot v12.5 (Systat Software Inc., San Jose, CA, USA) with a two-way ANOVA to calculate for significant differences between genotype and age, and with GraphPad Prism v8 applying multiple *t*-tests for testing within one age group.

## Figures and Tables

**Figure 1 ijms-22-07062-f001:**
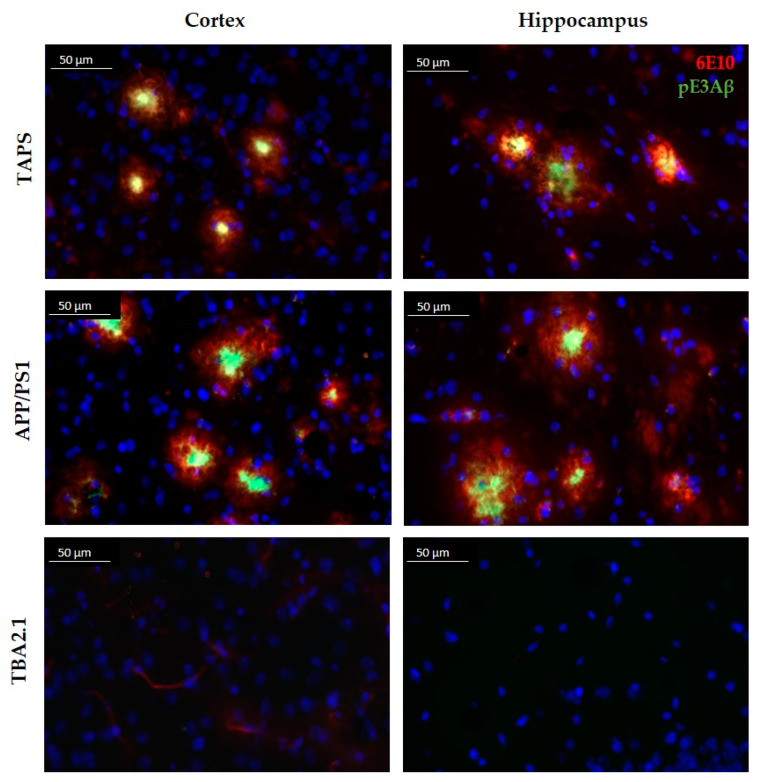
Comparison of the plaque morphology in TAPS, APP/PS1, and TBA2.1 mice. Sagittal slices stained with 6E10 (red) for whole amyloid-β (Aβ), and pE3 antibody (green) for truncated pyroglutamate Aβ (pE3Aβ) species at 400× magnification. Intense labeling of pE3Aβ in the center core of Aβ plaques was visible in the cortex of TAPS and APP/PS1, as well as in the hippocampus (CA3 region). No Aβ accumulation was visible in the cortex and hippocampus of heterozygous TBA2.1 mice. Cell nuclei labeled by DAPI (blue). Images of representative animals are shown; for total animal numbers, see Table 1.

**Figure 2 ijms-22-07062-f002:**
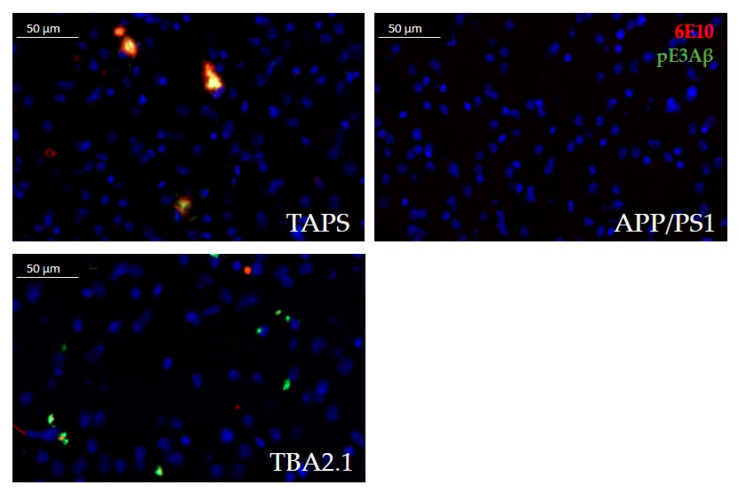
Comparison of Aβ accumulation in the lateral striatum at 400× magnification. TAPS mice accumulated Aβ aggregates in the lateral striatum, positive for total Aβ (red, 6E10) and pE3Aβ (green, pE3). In TBA2.1 mice, Aβ deposits were smaller but also positive for total Aβ and intensively stained for pE3Aβ. APP/PS1 mice in contrast were lacking any Aβ accumulation in this area, showing only cell nuclei, positive for DAPI (blue). Images of representative animals are shown; for total animal numbers, see Table 1.

**Figure 3 ijms-22-07062-f003:**
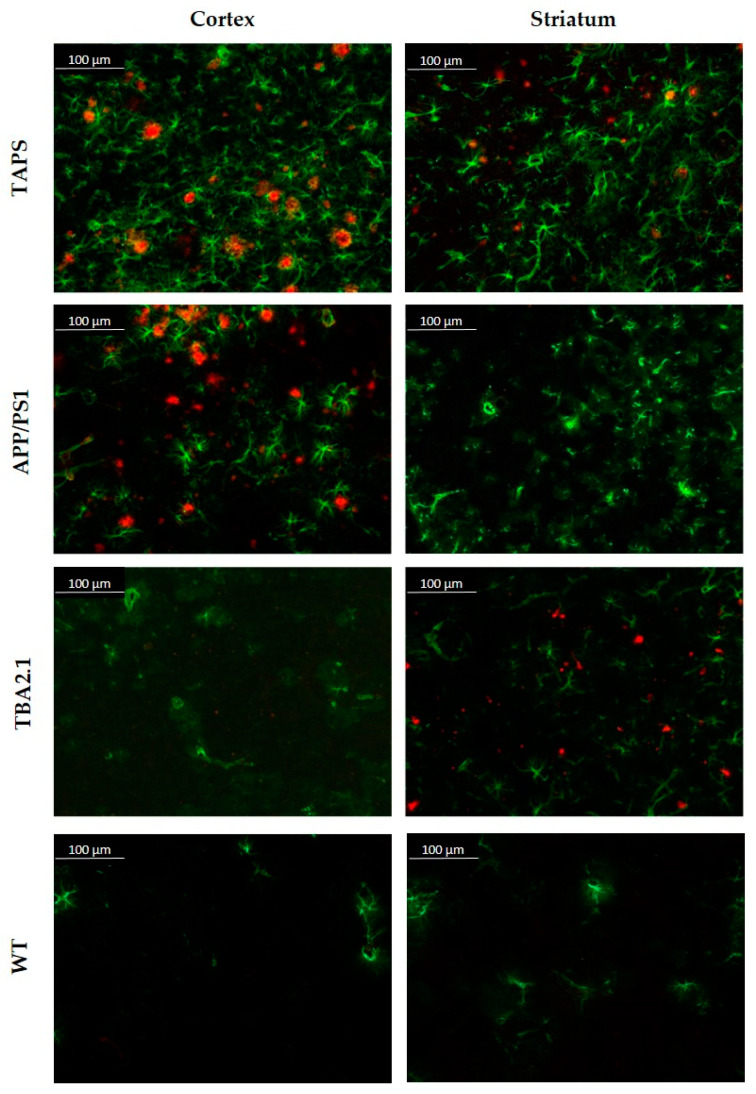
Amyloid-β plaque pattern and reactive astrogliosis in TAPS, APP/PS1, TBA2.1, and wild-type (WT) mice at 200x magnification. Immunofluorescence analysis of Aβ (red, 6E10) and reactive astrocytes (green, GFAP). Both, TAPS and APP/PS1 mice showed abundant plaque load throughout the whole cerebral cortex, accompanied by strong reactive astrogliosis. TBA2.1 mice seemed to lack such aggregation of Aβ in this area. Reactive astrogliosis in the cortex was on comparable low levels, as observed in WT mice. In the striatum, only TAPS and TBA2.1 mice showed higher amounts of Aβ aggregates, which could not be observed in APP/PS1 mice. Images of representative animals are shown; for total animal numbers, see Table 1.

**Figure 4 ijms-22-07062-f004:**
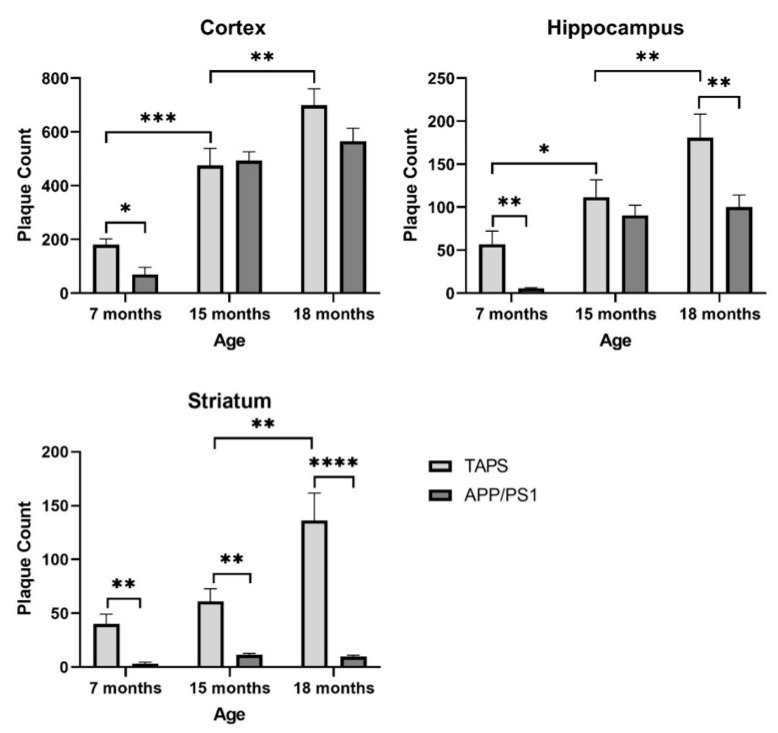
Plaque quantification in the brain of TAPS and APP/PS1 mice. The number of plaques was quantified on brain sections after immunostaining against total Aβ. TAPS mice showed an earlier onset in plaque formation in all investigated areas than APP/PS1 and a stronger progression until the age of 18 months. For statistical analysis, a two-way ANOVA was used to compare the age related, as well as a multiple *t*-test to analyze for genotype-dependent differences. Total number of animals was TAPS (*n* = 16), APP/PS1 (*n* = 15). For exact animal numbers per age, see Table 1. Data are given as mean + SEM; * *p* < 0.05; ** *p* < 0.01; *** *p* < 0.001; **** *p* < 0.0001.

**Figure 5 ijms-22-07062-f005:**
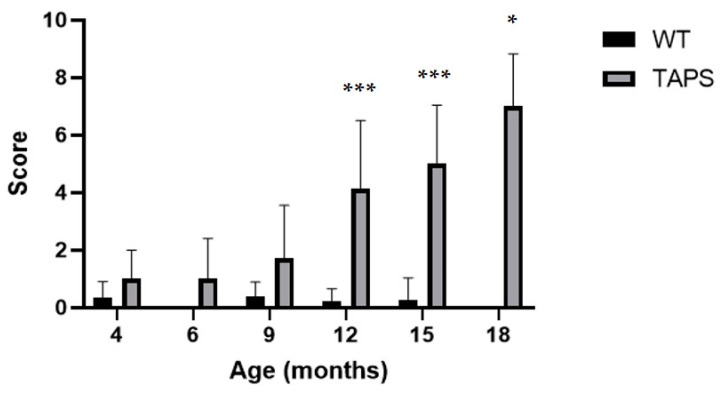
TAPS mice had age-dependent phenotypic alterations in the SHIRPA test. Shown are the mean scores of animals from each age group with SEM for TAPS and wild-type (WT) mice. A significant increase of SHIRPA scores was found for 12- (*** *p* = 0.0001, TAPS *n* = 16, WT *n* = 9), 15- (*** *p* = 0.0002, TAPS *n* = 11, WT *n* = 7), and 18- (* *p* = 0.0273, TAPS *n* = 4, WT *n* = 3)-month-old TAPS compared to their WT littermates. Scores of WT remained widely unchanged over time. For the statistical analysis, mixed effects analysis with Sidak´s post-hoc was used to compare the age and genotype. For all animal numbers per age, see Table 1. Data is given as mean + SEM.

**Figure 6 ijms-22-07062-f006:**
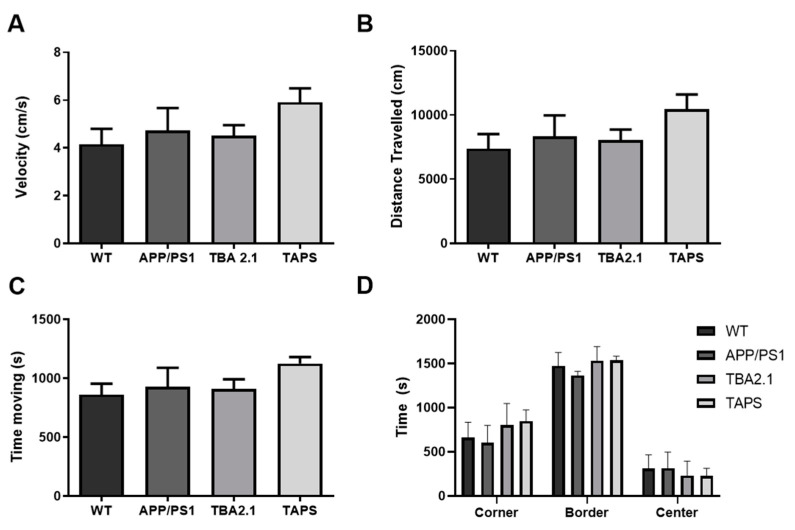
Performance of TAPS mice in the open field test. Wild-type (WT; *n* = 8), APP/PS1 (*n* = 5), and TBA2.1 (*n* = 8) showed similar velocity (**A**), distance traveled (**B**), and activity time (**C**) in the open field test. The TAPS mice (*n* = 4) showed a trend to be faster, traveled more, and were more active compared to the WT. All mice stayed similar time in the border, center, and corner regions of the open field (**D**). Data is given as mean + SEM.

**Figure 7 ijms-22-07062-f007:**
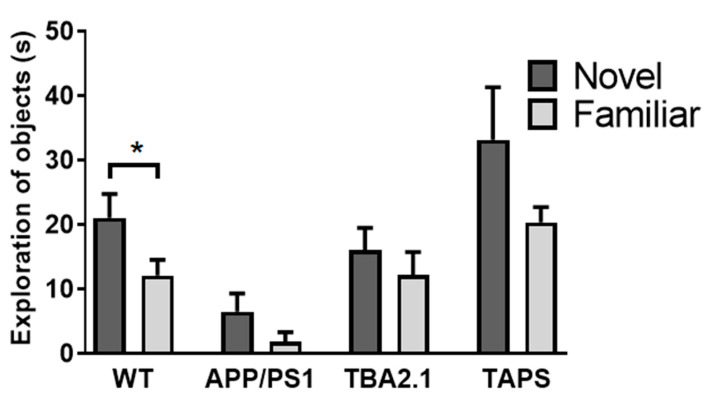
Deficits in the novel object recognition test in TAPS mice. The time animals explored the novel in comparison to a familiar object is given for each mouse line. Only wild-type mice (WT; *n* = 8) were able to discriminate between the novel and the familiar object (* *p* = 0.0380) while APP/PS1 (*n* = 5), TBA2.1 (*n* = 8), and TAPS (*n* = 4) mice showed a cognitive deficit in this task. For statistical analysis, a paired t-test was used to evaluate the difference between the exploration time of the novel and familiar object. Data is given as mean + SEM.

**Figure 8 ijms-22-07062-f008:**
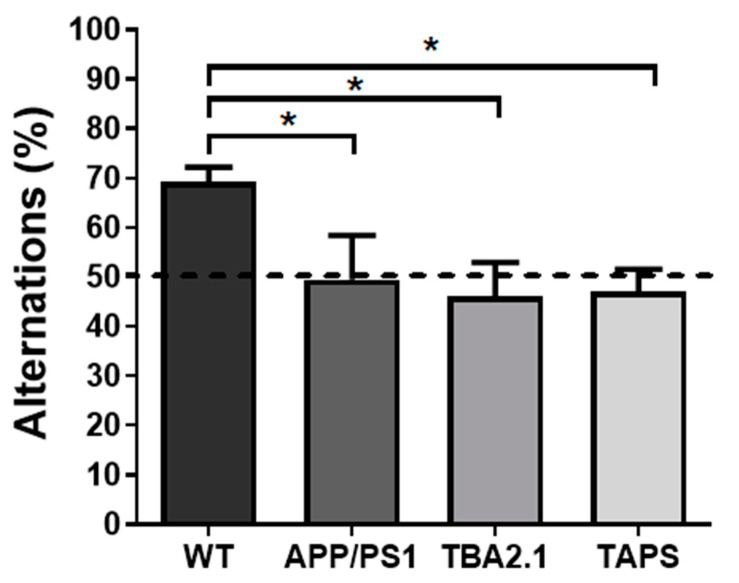
Deficit in spontaneous alternation of TBA2.1, APP/PS1, and TAPS mice in the T-maze. The spontaneous alternation was calculated as the ratio of entries into the correct arm to the amount of total trials in the T maze. With 20 months of age, the APP/PS1 (* *p* = 0.0349; *n* = 6), TAPS (* *p* = 0.0349; *n* = 6), and TBA2.1 (* *p* = 0.0145; *n* = 10) mice alternated significantly less than the wild-type (WT; *n* = 10) mice. Dashed line indicates threshold of chance (50%). For statistical analysis, one-way ANOVA with Holm-Sidak’s post-hoc was used to compare the genotypes. Data is given as mean + SEM.

**Figure 9 ijms-22-07062-f009:**
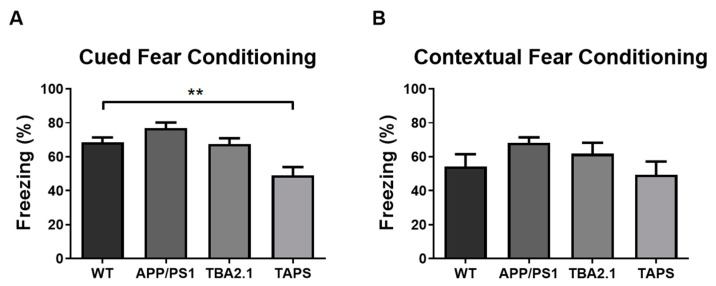
Impaired fear conditioning learning in TAPS mice. The percentage of freezing behavior was measured in the contextual and cued fear conditioning paradigm with 18 months of age. In the cued fear conditioning (**A**), the TAPS mice (*n* = 4) froze less compared to the wild-type (WT; *n* = 8) mice indicating a deficit in fear conditioning learning (** *p* = 0.0042). Both, APP/PS1 (*n* = 4) and TBA2.1 (*n* = 8), showed similar freezing behavior compared to WT mice, indicating intact fear conditioning learning. In the contextual fear conditioning (**B**), all genotypes showed similar freezing behavior. For statistical analysis, one-way ANOVA with Dunnettt´s post-hoc was used to compare the genotypes. Data is given as mean + SEM.

**Figure 10 ijms-22-07062-f010:**
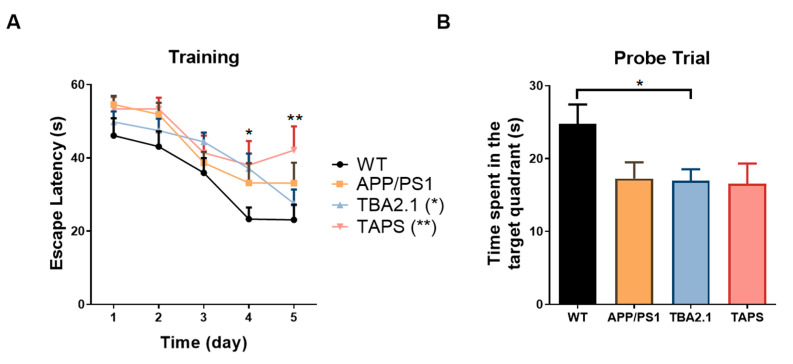
Performance in the Morris water maze with 20 months of age. In the training trials (**A**), the TAPS mice (*n* = 6) took longer to find the hidden platform compared to the wild-type mice (WT; *n* = 11) on the last day (** *p* = 0.0087), as well as TBA2.1 (*n* = 12) on the fourth day (* *p* = 0.0241), indicating a spatial learning deficit. In the probe trial (**B**), the TBA2.1 mice spent significantly less time in the target quadrant compared to WT mice (* *p* = 0.0319), while APP/PS1 (*n* = 8) and TAPS mice showed a non-significant trend towards reduced memory retrieval as they spent less time in the target quadrant than the WT mice. For statistical analysis, two-way and one-way ANOVA with Dunnett´s post-hoc were used to compare the genotypes during the training and in the probe trial, respectively. Data are given as mean + SEM.

**Table 1 ijms-22-07062-t001:** Number of mice used for each analysis according to genotype and age. MWM, Morris water maze; NOR, novel object recognition test; WT, wild-type.

Analysis		Age	Number of Mice/Genotype			
			WT	TAPS	APP/PS1	TBA 2.1
Histology		6.4 ± 0.3	-	6	5	3
		9.2 ± 0.4	-	1	3	3
		15.1 ± 0.4	-	5	3	-
		18.0 ± 0.4	-	4	4	1
		24.8 ± 1.3	5	6	5	5
SHIRPA		4.1 ± 0.2	3	3	-	-
		5.5 ± 0.4	4	8	-	-
		8.6 ± 0.4	8	18	-	-
		12.5 ± 0.3	9	16	-	-
		14.9 ± 0.2	7	11	-	-
		17.6 ± 0.6	3	4	-	-
Open Field		18.8 ± 0.7	8	4	5	8
NOR		18.8 ± 0.7	8	4	5	8
T-maze		18.4 ± 0.7	13	9	7	13
		20.7 ± 0.7	10	6	6	10
Fear Conditioning	Cued	18.8 ± 0.7	8	4	4	8
	Contextual	18.8 ± 0.7	8	4	4	8
		20.8 ± 0.7	7	2	3	6
MWM		20.7 ± 0.7	11	6	8	12

## Data Availability

All data from the study is available in this Manuscript.

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
