# Peer review of "PEAβ Triggers Cognitive Decline and Amyloid Burden in a Novel Mouse Model of Alzheimer’s Disease"

_ijms, 2021, doi:10.3390/ijms22137062_

Round 1
Reviewer 1 Report
Camargo et al. report a novel mouse model (TAPS) that mimics the pathophysiological events of AD using the pyroglutamate Abeta. The authors present a thorough analysis of the new model and a convincing dataset. A few minor issues remain.
- L18: “(data not shown)”. This data should be presented in the supplement or a reference should be used if such data was presented previously. Same for data in L255 and L293.
- Open Field test: Was the test repeated multiple times with the same animal? If so, was there any difference in their behavior between the initial and repeat tests? Also, did the authors analyze rearing behavior?
- Was the difference in exploration time significant between the WT and TAPS mice (Fig. 7)? Please explain why this may be.
Minor issues:
- Manuscript needs another round of proofreading since few grammatical errors have evaded the authors, e.g. L65: “neuroinflammation with six months” > “neuroinflammation within six months”
- Scale on figures should be readable.
- Fig. 2 the mouse designations are not clearly visible, only the very top of the white text is visible. Fig. 5 has similar issues with asterisk. Though this may be an artifact in my Acrobat. Please double check.
- Statistical comparison should mention the method used so readers do not need to dig in the methods to find it, e.g. in L193-5. Same in Fig. 4 and other places in the text that uses statistical comparison.
Author Response
Dear Reviewer 1,
We thank you for your valuable comments and suggestions, which have led to an improved version of the manuscript. We are confident that the revised version is now acceptable for publication and look forward to your decision.
Best regards,
Antje Willuweit
Reviewer 1
Camargo et al. report a novel mouse model (TAPS) that mimics the pathophysiological events of AD using the pyroglutamate Abeta. The authors present a thorough analysis of the new model and a convincing dataset. A few minor issues remain.
L18: “(data not shown)”. This data should be presented in the supplement or a reference should be used if such data was presented previously. Same for data in L255 and L293.
Answer: As requested, all data is now placed in the supplemental material and it is referred in the corresponding place in the manuscript, as shown below.
Lines 116-118: “TAPS mice were viable and fertile but showed a 14% increased rate of premature death in comparison to wild type (WT) littermates. For comparison, APP/PS1 mice showed a 3 % increased rate of premature death within this colony (table S1).”
Lines 269-271: “Analysis of the bodyweight over time demonstrated that overall there was no genotype dependent discrepancy observable for both tested genders (table S2).”
Lines 313 -314: “With 18 months of age all groups had similar amounts of alternations (Figure S2)”.
Open Field test: Was the test repeated multiple times with the same animal? If so, was there any difference in their behavior between the initial and repeat tests? Also, did the authors analyze rearing behavior?
Answer: We were not able to analyze rearing behavior, since this is not possible with our video setup. The Open Field test was done once at 18 months of age and not at 21 months of age. This information is located in the Materials and Methods sections (section 4.1 and table 1). However, in order to make this point clearer, we included this information additionally into the results part.
Lines 280- 281: “In the Open Field test, which was done once at 18 months, APP/PS1 and TBA2.1 did not differ significantly from the WT mice.”
Was the difference in exploration time significant between the WT and TAPS mice (Fig. 7)? Please explain why this may be.
Answer: There was no significant differences in the exploration time of the novel object between the WT and TAPS, though the reviewer is right, it seems to have a tendency. For the statistical analysis, the unpaired Students T-test was used (p = 0.1399) regarding the novel object exploration only. The same analysis was applied regarding the total exploration (novel + familiar object,) p = 0.46. In order to avoid misunderstanding, we have rearranged the following paragraph in the results section of the manuscript:
Lines 298 – 303: “For all other lines exploration times between novel and familiar object did not reach statistical significance, although there is some trend for more exploration of the novel object (Figure 7). High variability and low animal numbers, and low overall exploration may be accounted for this in case of the TAPS and APP/PS1 line, respectively. Regarding total exploration time of the objects, TAPS mice showed a tendency towards higher exploration of both objects, however, this did not reach statistical significance.”
-Manuscript needs another round of proofreading since few grammatical errors have evaded the authors, e.g. L65: “neuroinflammation with six months” > “neuroinflammation within six months”
Answer: Thank you for thoroughly reading the manuscript, we have checked the manuscript once again with special emphasis on grammatical errors.
- Scale on figures should be readable.
Answer: Thank you for this remark. In the revised manuscript all scale bars have been replaced to improve readability and are referred to in the legend.
- Fig. 2 the mouse designations are not clearly visible, only the very top of the white text is visible. Fig. 5 has similar issues with asterisk. Though this may be an artifact in my Acrobat. Please double check.
Answer: All alterations were made in the corresponding figures.
- Statistical comparison should mention the method used so readers do not need to dig in the methods to find it, e.g. in L193-5. Same in Fig. 4 and other places in the text that uses statistical comparison.
Answer: The reviewer has a point; all statistical analysis was now placed together with the p value in both text and figure cation. E.g.
Line 255-256: “The latter was highly significant compared to the values of the WT littermates (mixed effects analysis, genotype p < 0.0001, age p < 0.0001, interaction p < 0.0001; Sidak’s post-hoc test 12 m, p = 0.0001).

Reviewer 2 Report
This manuscript describes the histological and behavioural characterization of novel mouse model (TAPS mice) of Alzheimer’s disease (AD) that develops neuritic plaques containing pyroglutamate modified amyloid-β (pEAβ) species in the brain. The TAPS line was developed by intercrossing of the pEAβ-producing TBA2.1 mice, which do not develop neuritic plaques, with the plaque developing line APPswe/PS1dE9 bringing human APP and PS1 mutations. Authors claim that in TAPS mice amyloid pathology is accelerated with earlier onset and increased deposition of neuritic plaques in the brain compared to the other transgenic mice. Furthermore, Authors claim that the TAPS mice displayed a faster and more pronounced cognitive decline in comparison to the parental lines. Indeed, at the age of 7 months, TAPS mice showed higher levels of Aβ in cortex, compared to APP/PS1 mice. The behavioural findings are much less clear. I have the following points that need clarification:
- The results of the behavioural tests are quite confused. In general, it is surprising that behavioural deficit of the three transgenic AD mice were detected only at 18-20 months of age, an advance age that make their practical use quite inconvenient.
- The hyperactivity showed by the TAPS animals is quite unusual and has unclear pharmacological significance. All the three type of AD transgenic mice failed to show significant differences compared to wild-type animals in the contextual fear condition (Figure 9, right panel). In the Cued Fear Conditioning test a small but significant difference was detected between TAPS mice and wild-type mice, although the pharmacological relevance of this difference is unclear.
- In the Novel Object Recognition test, formally the observation time of novel and familiar objects in TAPS mice at 18 months of age did not differ significantly. However, there was a clear trend in favour of the novel object (Figure 7). Apparently, the difference did not reach statistical significance because average exploration time was higher in TAPS mice and there was much higher variability compared to wild-type mice.
- In the Morris water maze test performed at 20 months of age, the mean time spent in the target quadrant of APP/PS1, TBA2.1 and TAPS mice was basically identical, but only the mean value of the TBA2.1 mice was significantly different compared to that of wild-type mice. (Figure 10, Probe Trial). I have the impression that the number of animals tested was too small to detect statistical significant differences compared to control mice.
- Surprisingly, no assessments of tau deposition and, importantly, neurodegeneration were performed or presented by the Authors. Thus, we do not know if the new TAPS mice model affect neurodegeneration, which is a typical feature of human AD. The same applies for tau and phosphor-tau pathology.
- TAPS mice displayed neuritic plaques in the striatum but this pathology has un unclear pharmacological meaning since autopsy studies suggest that striatal amyloid plaques may be mainly restricted to subjects in higher Braak neurofibrillary tangle stages.
- In the Introduction (page 2), “familial AD” is misspelled.
Author Response
Dear Reviewer 2,
We thank you for your valuable comments and suggestions, which have led to an improved version of the manuscript. We are confident that the revised version is now acceptable for publication and look forward to your decision.
Best regards,
Antje Willuweit
Reviewer 2
This manuscript describes the histological and behavioural characterization of novel mouse model (TAPS mice) of Alzheimer’s disease (AD) that develops neuritic plaques containing pyroglutamate modified amyloid-β (pEAβ) species in the brain. The TAPS line was developed by intercrossing of the pEAβ-producing TBA2.1 mice, which do not develop neuritic plaques, with the plaque developing line APPswe/PS1dE9 bringing human APP and PS1 mutations. Authors claim that in TAPS mice amyloid pathology is accelerated with earlier onset and increased deposition of neuritic plaques in the brain compared to the other transgenic mice. Furthermore, Authors claim that the TAPS mice displayed a faster and more pronounced cognitive decline in comparison to the parental lines. Indeed, at the age of 7 months, TAPS mice showed higher levels of Aβ in cortex, compared to APP/PS1 mice. The behavioural findings are much less clear. I have the following points that need clarification:
1. The results of the behavioural tests are quite confused. In general, it is surprising that behavioural deficit of the three transgenic AD mice were detected only at 18-20 months of age, an advance age that make their practical use quite inconvenient.
Answer: Since it is already published that the TBA2.1 mice do not develop any deficits before 20 months and we have observed that, in our work group, the APP/PS1 also develop deficits very late (after 20 months), we have decided to start all experiments with 18 months. In order to clarify this, we modified the methods and discussion sections in the manuscript.
Materials and methods, lines 595 – 598: “Based on the late behavioral alterations the parental lines displayed in our hands in the past, the cognitive test battery was started with 18 months of age. The first cohort of TAPS and WT littermates was tested in the SHIRPA test battery for initial check of behavioral abnormalities.”
Discussion, lines 536-543: “In general, the behavioral deficits of all three mouse lines, and especially the APP/PS1 and TAPS mice, appeared with an advanced age of 18 to 20 months, which might limit their practical use for future studies. On the other hand, mouse models with very aggressive phenotype progression and early behavioral deficits before clear pathological alterations have been critisized, because their relevance to the clinical disease has been questioned [68]. In this respect the new TAPS line joins the ranks of the rather late AD models in which the cognitive deficits clearly develop as a consequence of pathological processes.
2. The hyperactivity showed by the TAPS animals is quite unusual and has unclear pharmacological significance. All the three type of AD transgenic mice failed to show significant differences compared to wild-type animals in the contextual fear condition (Figure 9, right panel). In the Cued Fear Conditioning test a small but significant difference was detected between TAPS mice and wild-type mice, although the pharmacological relevance of this difference is unclear.
Answer: Although it is not so common, hyperactivity was observed in some transgenic models that have a disturbance in the basal ganglia network. Since TAPS mice have Aβ particles in the striatum, one might speculate that there is an alteration in the circuitry due to Aβ induced toxicity. Regarding the fear conditioning, it is known that in AD patients the fear memory is also impaired. Therefore, the new TAPS model shows the classical cognitive deficits as well as additional impairments observed in AD patients. In order to explain the relevance of both hyperactivity and cued fear conditioning impairment, we have added the following parts to the discussion:
Lines 496 – 508: “In addition, we observed a trend towards higher velocity, longer distance travelled and generally more activity in the Open Field test, indicating a hyperactive phenotype of TAPS mice compared to WT and TBA2.1 mice. One might speculate, whether these motor alterations could be due to the development of Aβ pathology and neuroinflammation in the striatum. The striatum, as a key interface for excitatory and inhibitory neurons, plays a major role in action selection and motor function [58]. Moreover, in other AD mouse models the observed hyperactivity was correlated with basal ganglia circuitry dysfunction [59-61], therefore, one might assume that striatal Aβ deposits induce changes in the basal ganglia network. So far, the relevance of a hyperactive phenotype in AD mouse models is not quite clear. Although hyperactivity is also part of the behavioral and psychological symptoms of dementia in patients, its causes are still under discussion and have not been sufficiently investigated yet (for review see [62])”
Lines 473 – 477: “Finally, the Cued Fear Conditioning paradigm is based on fear memory with involvement of the amygdala [50,51], a brain region which was also affected by Aβ pathology in TAPS mice. Deficits in fear memory, particularly in the Cued Fear Conditioning, have also been observed in AD patients [52,53] and other transgenic AD models before [54,55].“
3. In the Novel Object Recognition test, formally the observation time of novel and familiar objects in TAPS mice at 18 months of age did not differ significantly. However, there was a clear trend in favour of the novel object (Figure 7). Apparently, the difference did not reach statistical significance because average exploration time was higher in TAPS mice and there was much higher variability compared to wild-type mice.
Answer: The reviewer is absolutely right, since the variation is high, we were not able to see a significant difference, even though there is a trend. The following changes were made accordingly, in the results and discussion section of the manuscript.
Results, line 298-301: “For all other lines exploration times between novel and familiar object did not reach statistical significance, although there is some trend for more exploration of the novel object (Figure 7). High variability and low animal numbers, and low overall exploration may be accounted for this in case of the TAPS and APP/PS1 line, respectively”.
Discussion, line 458-459: “TAPS mice developed cognitive deficits beginning with 18 months of age mainly in the Cued Fear Conditioning.”
Discussion, line 478-479: “Even though TBA2.1 mice do not develop Aβ plaques, they showed clear deficits in both the T-maze and the MWM.”
4. In the Morris water maze test performed at 20 months of age, the mean time spent in the target quadrant of APP/PS1, TBA2.1 and TAPS mice was basically identical, but only the mean value of the TBA2.1 mice was significantly different compared to that of wild-type mice. (Figure 10, Probe Trial). I have the impression that the number of animals tested was too small to detect statistical significant differences compared to control mice.
Answer: As it was already specified in the Morris Water Maze results sections, we can observe a non-significant trend that TAPS and APP/PS1 explored less the target quadrant in the probe trial. In order to make this clearer, we have included the following in the manuscript.
Lines 360 – 363: “In the probe trial, both TAPS and APP/PS1 mice had a tendency towards reduced searching time in the target quadrant compared to WT but this difference did not reach statistical significance, presumably due to insufficient animal numbers.”
5. Surprisingly, no assessments of tau deposition and, importantly, neurodegeneration were performed or presented by the Authors. Thus, we do not know if the new TAPS mice model affect neurodegeneration, which is a typical feature of human AD. The same applies for tau and phosphor-tau pathology.
Answer: Considering that none of the transgenic amyloidosis models do develop tau deposition and some display only very limited tau phosphorylation [68], we did not evaluate this in the current study. However, we added this aspect into the discussion of the manuscript.
Lines 532 – 536: “Finally, the new TAPS line is an amyloidosis model that reflects several aspects of the human disease. However, the aspect of tauopathy, which is an important feature of human AD, is missing. Since none of the previously described amyloidosis models shows excessive tau phosphorylation and also no neurofibrillary tangles [68], development of tauopathy in TAPS mice was not to be expected.”
Regarding the neurodegeneration, the reviewer has a point; therefore, we have performed an additional analysis and evaluated the CA1 region of the hippocampus for loss of neurons. This analysis is now included as figure S1 in the supplement, in the results and in the discussion section of the manuscript.
Materials and Methods, lines 643 – 649 :”For analysis of neuron loss, a subset of six slides per animal was analyzed from 24 months old TAPS and WT mice.
After fixation in 4% buffered Formalin, slides were stained for 5 min. in 5 μg/mL DAPI solution (4′,6-Diamidin-2-phenylindol, Sigma Aldrich, Steinheim, Germany). Microscope images of the hippocampus were taken at 50x magnification with the Leica LMD 6000 Fluorescent Microscope. CA1 region of the hippocampus was evaluated for the number of positive nuclei via Cell Profiler Software (Broad Institute, Cambridge, USA).
Results, lines 188 – 193: “Since neuron loss in the CA1 region of the hippocampus was described for homozygous TBA2.1 mice [28] , we analyzed this brain region also in the TAPS line. Quantification of neuronal nuclei in the hippocampus of 24 months old TAPS and WT mice showed no significant differences in neuronal density in the stratum pyramidale of the CA1. Cell counts were on comparable levels, therefore no detectable signs of neurodegeneration could be observed in the designated area for TAPS compared to WT mice (Figure S1).”
Discussion, lines 449 – 457: “Despite showing a strong reactive astrogliosis and an increased inflammatory milieu in the investigated brain areas, together with a high plaque load, however, neurodegeneration could not be observed in the hippocampus of TAPS mice, compared to WT. The CA1 region of the hippocampus, in particular the stratum pyramidale showed comparable amounts of neurons throughout the investigated area of TAPS and WT mice. Although neurodegeneration in this brain region was described for homozygous TBA2.1 mice [28], obviously, the heterozygous status of the TBA2.1 transgene was not sufficient to induce neuron loss in TAPS mice. However, this does not exclude the presence of subtle neurodegenerative changes throughout the entire brain, which we might have overlooked. “
6. TAPS mice displayed neuritic plaques in the striatum but this pathology has un unclear pharmacological meaning since autopsy studies suggest that striatal amyloid plaques may be mainly restricted to subjects in higher Braak neurofibrillary tangle stages.
Answer: Regarding the role of striatal Aβ, PiB-PET measurements detected striatal amyloid at the preclinical stages of AD [35] and were more closely correlated with tau pathology as well as memory deficits, therefore, it was discussed as another predictor of AD at the preclinical stage [36]. Certainly, all animal models have methodological limitations. The TAPS line was developed as an additional tool in the AD research, which could provide a better understanding of non-cognitive, combined with the cognitive symptoms of AD patients and elucidate further the role of pEAbeta. All in all, the reviewer has a valid point; we have added this aspect to the discussion in order to be more complete regarding the model limitation.
Lines 399-410: “Most striking and in addition to what can be observed in the APP/PS1 line, TAPS mice showed also Aβ aggregates in the lateral striatum, which appeared larger and more mature than the small Aβ particles present in the striatum of the parental TBA2.1 line. We could confirm data by Alexandru et al. [28] that heterozygous TBA2.1 mice at 21 months had only a low amount of N-terminally truncated and pyroglutamate-modified Aβ in small intracellular aggregates in the striatum which were, however, not sufficient to induce motor deficits in this line. The clinical relevance of Aβ aggregates in the striatum are not yet clear, although striatal amyloid plaques have also been found in AD patients. Striatal Aβ depositions have been described mainly in AD patients of advanced stages [31-34], but may also occur in the preclinical stage [35]. A recent study showed that Aβ deposition in the striatum correlates with both, memory deficits and tau pathology [36].”
7. In the Introduction (page 2), “familial AD” is misspelled.
Answer: we have changed it to the correct spelling.

Round 2
Reviewer 2 Report
I believe the revised manuscript is significantly improved.
Author Response
Dear Reviewer,
thank you very much for your valuable suggestions which led to an improvement of the manuscript.
sincerely, Antje Willuweit